# NoWag: A Unified Framework for Shape Preserving Compression of Large Language Models

**Lawrence Liu**[1]   **Inesh Chakrabarti**[1]   **Yixiao Li**[2]   **Mengdi Wang**[3]   **Tuo Zhao**[2]
**Lin F. Yang**[1]

[1]*University of California, Los Angeles* [2]*Georgia Institute of Technology* [3]*Princeton University*
{lawrencerliu, inesh33}@ucla.edu, yixiaoli@gatech.edu
mengdiw@princeton.edu, tourzhao@gatech.edu, linyang@ee.ucla.edu

## Abstract

Large language models (LLMs) exhibit remarkable performance across various natural language processing tasks but suffer from immense computational and memory demands, limiting their deployment in resource-constrained environments. To address this challenge, we propose NoWag (**No**rmalized **W**eight and **A**ctivation **G**uided Compression), a unified framework for one-shot shape preserving compression algorithms. We apply NoWag to compress Llama-2 (7B, 13B, 70B) and Llama-3 (8B, 70B) models using two popular shape-preserving techniques: vector quantization (NoWag-VQ) and unstructured/semi-structured pruning (NoWag-P). Our results show that NoWag-VQ significantly outperforms state-of-the-art one-shot vector quantization methods, while NoWag-P performs competitively against leading pruning techniques. These findings highlight underlying commonalities between these compression paradigms and suggest promising directions for future research. Our code is available at https://github.com/LawrenceRLiu/NoWag

## 1 Introduction

Large language models (LLMs) (Brown et al., 2020) have demonstrated remarkable capabilities across a wide range of fields and tasks (Huang & Yang, 2025; Wei et al., 2022; Park et al., 2023), but their immense computational and memory requirements during inference pose significant challenges for deployment. Consequently, post-training compression techniques have emerged as a promising tool to reduce model size and computational overhead while maintaining accuracy. Two promising families of methods for post-training compression are Pruning (LeCun et al., 1989; Hassibi et al., 1993; Han et al., 2015) and Quantization (Yao et al., 2022; Dettmers et al., 2022; Ahmadian et al., 2023; Li et al., 2025).

Pruning aims to remove redundant parameters from LLMs while preserving performance. We will focus on two forms of pruning, unstructured pruning (Liao et al., 2023), which removes/zeros out individual elements without any structure, and N:M semi-structured pruning (Huang et al., 2025), where strictly N of every M elements are zeroed out. SparseGPT (Frantar & Alistarh, 2023) introduced an efficient, unstructured and semi-structured pruning method that leverages Hessian-based weight updates to minimize performance loss. More recently, Wanda (Sun et al., 2023) demonstrated a simple yet effective unstructured and semi-structured pruning method that requires no weight updates or hessian computation, making it significantly faster and easier to apply than SparseGPT. However, current hardware only supports 2:4 semi-structured sparsity, which leads to significant performance degradation after compression.

A more effective compression method is quantization, which reduces the number of bits used to store each weight (Kuzmin et al., 2023). In this paper, we focus on weight-only Post-Training Quantization (PTQ), a common form of quantization. Pioneering works (Frantar et al., 2022; Lin et al., 2024; Kim et al., 2023) focused on scalar quantization. For extreme compression (e.g., $\leq 4$ bits per weight), Vector Quantization (VQ), where groups

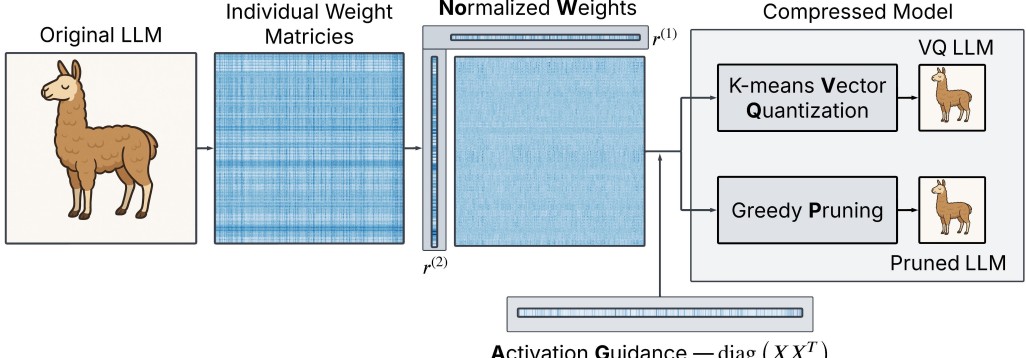

Figure 1: Illustration of proposed NoWag (**No**rmalized **W**eight and **A**ctivation Guided Compression). Given a LLM, compression of each weight matrix W is performed independently. Vectors $r^{(1)}$ and $r^{(2)}$ are used to normalize W. With the second moment of the activations $\mathrm{diag}\left(XX^T\right)$ guiding the importance of each weight for the compression algorithm, such as K-means VQ (NoWag-VQ) and Pruning (NoWag-P)

of $d$ consecutive weights are quantized together, has demonstrated superior performance because the codebook can be shaped to the distribution of weights (Egiazarian et al., 2024; Van Baalen et al., 2024; Tseng et al., 2024a; Liu et al., 2024). However, most current algorithms all share at least one of the following two drawbacks. First, an expensive weight-update process necessitating matrix inversion, similar to SparseGPT. Second, sampling a sufficiently accurate Hessian for quantization can require up to 25 million tokens and $\sim 1$ TB of CPU memory for a model like Llama-3 70B, introducing a new computational bottleneck.

In this work, we address these issues by formulating a unifying framework for shape-preserving compression algorithms, where the compressed weight matrix has the same shape as the original uncompressed counterpart but can be stored with less memory. This method is weight update free, less dependent on calibration data, and features a novel normalization technique that benefits both pruning and quantization. We term this family of compression methods NoWag (**No**rmalized **W**eight and **A**ctivation Guided Compression). We show that the VQ variation of NoWag, NoWag-VQ (**NoWag** for **V**ector **Q**uantization), outperforms the SOTA one-shot VQs QuIP# (Tseng et al., 2024a), at bits per value, while using 48x less calibration data. Furthermore, we show that the pruning variation of NoWag, NoWag-P (**NoWag** for **P**runing), outperforms Wanda, a SOTA pruning algorithm at retaining language modeling performance.

## 2 Problem Formulation

Given a trained LLM, our goal is to obtain a compressed model that significantly reduces the computational and memory requirements while retaining as much general performance as possible. Due to the large number of parameters, using global optimization for compression is computationally infeasible. As a result, a common approach is to independently perform data aware compression of each linear layer (Nagel et al., 2020), as well as to finetune the compressed LLM, (Egiazarian et al., 2024; Malinovskii et al., 2024; Tseng et al., 2024a) to recover performance.

More formally, the objective for shape preserving compression algorithms is: For a linear layer in an LLM with weight matrix $W \in \mathbb{R}^{d_{\mathrm{out}} \times d_{\mathrm{in}}}$, find a compressed weight matrix $\hat{W} \in \mathbb{R}^{d_{\mathrm{out}} \times d_{\mathrm{in}}}$ to replace W that requires less memory while minimizing the deviation from the original model's behavior.

In standard notation, given input activations $x \in \mathbb{R}^{d_{\text{in}}}$, the output is computed as $y = Wx$, where $y \in \mathbb{R}^{d_{\text{out}}}$. To incorporate data awareness, we sample $n$ sequences of length $l$ from a calibration dataset and collect the corresponding activation samples $X^T \in \mathbb{R}^{m \times d_{\text{in}}}$ where $m = n \times l$.

## 3   Related Works

Many existing works for shape preserving compression algorithms aim to minimize the expected error between the outputs of the original layer and the compressed layer (Nagel et al., 2020; Frantar & Alistarh, 2023; Frantar et al., 2022):

$$\ell(\hat{W}) = \mathbb{E}_x \left[ \|(W - \hat{W})x\|_2^2 \right] = \text{tr}(\left( (W - \hat{W}) \, \mathbb{E}_x[xx^T] \, (W - \hat{W})^T \right)) \tag{1}$$

Because the underlying distribution of $x$ is unknown, the expected value of the hessian $\mathbb{E}_x[xx^T]$ of the activations is estimated through the sample activations $\hat{H} = \frac{1}{m} XX^T \in \mathbb{R}^{d_{\text{in}} \times d_{\text{in}}}$, allowing for the following approximation, $\ell'$ of $\ell$:

$$\ell'(\hat{W}) = \text{tr}\left( (W - \hat{W}) \, \hat{H} \, (W - \hat{W})^T \right) \tag{2}$$

The challenge of minimizing such an objective is that it cannot be broken down into independent element wise subproblems. As a result, solving for the optimal pruned or quantized $\hat{W}$ is NP-hard. Even greedy approximations to these solution are themselves computationally expensive; for example, a common approach is a group wise greedy approach with linear feedback updating (Frantar et al., 2022; Tseng et al., 2024a; Liu et al., 2024; Van Baalen et al., 2024; Chee et al., 2023; Frantar & Alistarh, 2023). In such an approach, groups of rows are iteratively quantized/pruned in a greedy fashion, and the remaining non compressed rows are adjusted to compensate. Calculating the optimal compensation requires calculating the inverse of $\hat{H}$, which carries a computational overhead of $\mathcal{O}(d_{\text{in}}^3)$.

However, recent works appear to show that simplified approximations of equation 2 are sufficient, and even superior. For example, Wanda (Sun et al., 2023), only considers the diagonal elements of the sample hessian. This results in a pruning algorithm that prunes based on a score metric $S_{ij} = |W_{ij}| \|X_j\|_2$ without any feedback. Furthermore, for unstructured pruning, pruning is performed independently in per-output groups. In other words, instead of constraining each weight matrix, each individual row is constrained to $x\%$ sparsity. Although this constraint is more restrictive, Wanda reports that it results in superior performance competitive with SparseGPT. Likewise, for quantization, SqueezeLLM (Kim et al., 2023), replaces the Hessian with the diagonal of the Fischer information matrix, allowing for K-means to be used to cluster the weights without any computationally expensive weight updates. SqueezeLLM is able to outperform some linear feedback based scalar quantization algorithms such as GPTQ Frantar et al. (2022).

## 4   NoWag: (Normalized Weight and Activation Guided Compression)

**Compression Objective**   To ensure numerical stability and enhance compression efficiency, we first normalize $W$ as per Algorithm 1 to obtain $\bar{W} \in \mathbb{R}^{d_{\text{out}} \times d_{\text{in}}}$ using normalization vectors $r^{(1)} \in \mathbb{R}^{d_{\text{in}}}$ and $r^{(2)} \in \mathbb{R}^{d_{\text{out}}}$:

$$\bar{W}_{ij} = \frac{1}{r_i^{(2)}} \left( \frac{W_{ij}}{r_j^{(1)}} \right), \quad r_j^{(1)} = \sqrt{\sum_{i=1}^{d_{\text{out}}} W_{ij}^2}, \quad \forall j \in [d_{\text{in}}], \quad r_i^{(2)} = \sqrt{\sum_{j=1}^{d_{\text{in}}} \left( \frac{W_{ij}}{r_j^{(1)}} \right)^2}, \quad \forall i \in [d_{\text{out}}].$$

The compressed weight matrix $\hat{W}$ is obtained by minimizing the following weighted Frobenius norm:

$$\tilde{\ell}(\hat{W}) = \| \bar{W} - \hat{W} \|_{F,\text{diag}(XX^T)}^2 = \sum_i \sum_j (\bar{W}_{ij} - \hat{W}_{ij})^2 \|X_j\|_2^2. \tag{3}$$

**Algorithm 1** NoWag Normalization

**Require:** Weight matrix $W \in \mathbb{R}^{d_{\text{out}} \times d_{\text{in}}}$
**Ensure:** Normalized weight matrix $\bar{W}$, normalization vectors $r^{(1)}, r^{(2)}$

Compute column norms: $r_j^{(1)} = \sqrt{\sum_{i=1}^{d_{\text{out}}} W_{ij}^2}$ for $j \in [1, d_{\text{in}}]$

Normalize columns: $W'_{ij} = \frac{W_{ij}}{r_j^{(1)} + \varepsilon} \forall i, j$

Compute row norms: $r_i^{(2)} = \sqrt{\sum_{j=1}^{d_{\text{in}}} W_{ij}'^2}$ for $i \in [1, d_{\text{out}}]$

Normalize rows: $\bar{W}_{ij} = \frac{W'_{ij}}{r_i^{(2)} + \varepsilon} \forall i, j$

**return** $\bar{W}, r^{(1)}, r^{(2)}$

**Algorithm 2** NoWag Inference

**Require:** Input activation $x \in \mathbb{R}^{d_{\text{in}}}$, compressed weight matrix $\hat{W} \in \mathbb{R}^{d_{\text{out}} \times d_{\text{in}}}$, normalization vectors $r^{(1)} \in \mathbb{R}^{d_{\text{in}}}$, $r^{(2)} \in \mathbb{R}^{d_{\text{out}}}$
**Ensure:** Output activation $y \in \mathbb{R}^{d_{\text{out}}}$

Normalize input: $\tilde{x}_j \leftarrow x_j / r_j^{(1)}, \forall j \in [1, d_{\text{in}}]$

Compute intermediate output: $\tilde{y}_i \leftarrow \hat{W}_i x, \forall i \in [1, d_{\text{out}}]$

Denormalize output: $y_i \leftarrow \tilde{y}_i \cdot r_i^{(2)}, \forall i \in [1, d_{\text{out}}]$

**return** $y$

**Algorithm 3** NoWag-VQ

**Require:** Weight matrix $W \in \mathbb{R}^{d_{\text{out}} \times d_{\text{in}}}$, activation samples $X \in \mathbb{R}^{m \times d_{\text{in}}}$, subvector dimension $d$, target bits per value $n_{\text{bits}}$, number of iterations $n_{\text{iter}}$
**Ensure:** Quantized weight matrix $\hat{W} \in \mathbb{R}^{d_{\text{out}} \times d_{\text{in}}}$, codebook $C$, assignments $A$

$(r^{(1)}, r^{(2)}, \bar{W}) \leftarrow$ NoWag Normalization$(W)$ {See Algorithm 1}

$(\bar{W}_{\text{padded}}, d_{\text{in, padded}}) \leftarrow$ VQPadding$(\bar{W}, d_{\text{in}}, d)$ {See Algorithm 4}

$H_j \leftarrow \|X_j\|_2^2, \forall j \in [1, d_{\text{in}}]$ {Diagonal of sample hessian}

**if** $d_{\text{in}} \neq d_{\text{in, padded}}$ **then**
    Pad $H$ with zeros to length $d_{\text{in, padded}}$
**end if**

Reshape $\bar{W}_{\text{padded}}$ into subvectors: $\bar{W}_{\text{sub}} \in \mathbb{R}^{N \times d}$ where $N = \frac{d_{\text{out}} \times d_{\text{in, padded}}}{d}$

Reshape $H$ into subvector weights: $H_{\text{sub}} \in \mathbb{R}^{N \times d}$

$n_{\text{centroids}} \leftarrow 2^{n_{\text{bits}} \cdot d}$

$(C, A) \leftarrow$ Weighted_KMeans$(\bar{W}_{\text{sub}}, H_{\text{sub}}, n_{\text{centroids}}, n_{\text{iter}})$ {See Algorithm 6}

$\hat{W}_{\text{sub}} \leftarrow$ Map each subvector to its assigned centroid using $A$

Reshape $\hat{W}_{\text{sub}}$ back to matrix form and remove padding to get $\hat{W} \in \mathbb{R}^{d_{\text{out}} \times d_{\text{in}}}$ {Reverses padding from Algorithm 4}

**return** $\hat{W}, C, A$

Here, $X_j \in \mathbb{R}^m$ represents the calibration activations for the $j$th input channel, and $\|X_j\|_2^2$ acts as a weighting term that prioritizes important elements of $W$. A key benefit of such an objective is that it can be decomposed into independent subproblems at the element level, subject to global constraints imposed by the chosen compression algorithm. This allows for compression algorithms that do not require any feedback methods.

Denormalization can be done on the fly by multiplying the inputs by $r^{(2)}$ and the outputs by $r^{(1)}$ as noted in Algorithm 2. The additional computational cost of these operations scales linearly with the input and output dimensions respectively. They are therefore largely neglible compared with the overall computational cost of Matrix multiplication. Furthermore, even though these normalization vectors vectors are stored in fp16, the additional overhead is minimal: in terms of bits per value, they account for $< 0.01$ bits per value.

**Paradigms of Compression**     The above formulation unifies the following two paradigms of shape preserving compression.

1. **Quantization (NoWag-VQ):** With the discretization constraints imposed by quantization, optimizing equation 3 can be approximated by weighted K-means, where the weights are determined by $\text{diag}(XX^T)$. For initialization, a modified version of K-means++ was used, with the possible centroids sampled from a random subsample of the subvectors. The computational complexity of d dimensional K-means is $\mathcal{O}(dNTK)$ where $N$ is the number of data points, $K$ is the number of clusters, and $T$ is the number of iterations. Therefore the computational complexity of $d$ dimensional NoWag-VQ for quantization to $n_{\text{bpv}}$ bits per value is $\mathcal{O}(d_{\text{in}} d_{\text{out}} T 2^{n_{\text{bpv}} d})$. Note that in our experiments $T << \min(d_{\text{in}}, d_{\text{out}})$. The pseudo-code for this algorithm is detailed in Algorithm 3 and a detailed formulation of VQ can be found in Appendix A.

2. **Unstructured/Semi-Structured Pruning (NoWag-P):** For an $x$%-unstructured pruning pattern, where $x$% of the weight matrix entries $(i, j)$ are zeroed out, our method selects the $x$% of entries with the smallest $\bar{W}_{ij}^2 \|X_j\|_2^2$, thereby minimizing Equation 3. For N:M Semi-Structured Pruning, our method selects the N entries in each group of M with the smallest $\bar{W}_{ij}^2 \|X_j\|_2^2$. Through the quickselect algorithm

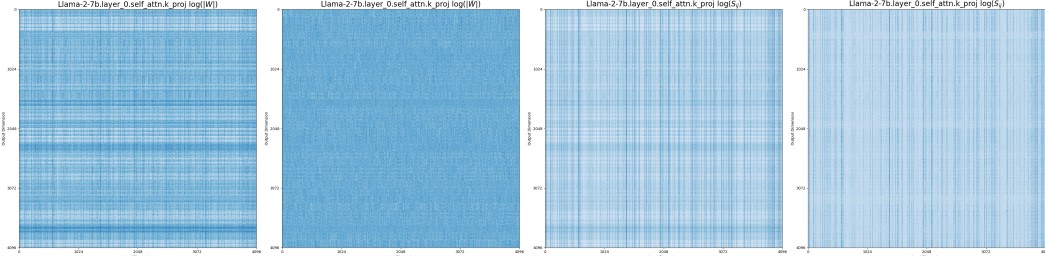

Figure 2: A sample weight from the first attention layer of Llama-2-7B. From left to right: visualization of the absolute values of the weights, normalized weights, importance scores, and normalized importance scores all down-sampled to 1:4 scale by max pooling. Individual elements are visualized in log scale, with blue implying larger value.

(Hoare, 1962), we can find the threshold for $x\%$ of the entries linear time on average (Mahmoud et al., 1995). Thus, NoWag-P has a computational complexity of $\mathcal{O}(d_{\text{in}}d_{\text{out}})$ on average. The pseudo code for NoWag-P is detailed in Algorithm 5 in Appendix F.

### 4.1 Why this works

The critical step in our approach is the normalization of $W$. Our normalization method effectively normalizes $W_{ij}$ by both the input and output group. This removes the biases on the compression algorithm to focus on smaller-magnitude rows/columns, leading to a better retention of the overall performance of an LLM.

To illustrate this, we visualize W and its normalized counterpart, $\bar{W}$ for an example weight matrix in figure 2. In addition, to understand the effects of data awareness, we also visualized the element wise importance scores $\bar{S}_{ij} = \|\bar{W}_{ij}\| \|X_j\|_2$ derived from equation 3. Likewise, for comparison, we also visualized the naive scores $S_{ij} = \|W_{ij}\| \|X_j\|_2$ without considering normalization.

We observe that non-normalized weights exhibit a structured pattern, with specific outlier rows and columns, with larger magnitudes. These structures can be attributed to several phenomenons, such as sensitive attention heads, rotary embedding patterns, and outlier features (Dettmers et al., 2024; Su et al., 2024; Dettmers et al., 2022; Vig, 2019; Olsson et al., 2022). In comparison, the normalized weights do not exhibit this patterns. This is highly beneficial for vector quantization, as it projects the $d$ dimensional distribution of consecutive weights into a bounded $[0, 1]$ "ball shaped" distribution, visualized in Figure 3

The importance visualizations in Figure 2 once again exhibits these row and column wise structures. Thus, when pruning is applied, the removed elements will be concentrated away from these rows and columns on the non-outlier columns. In turn, this effectively removes entire input/output channels, reducing the performance of the compressed LLM. Normalization largely removes the rowise outlier structures from the importance scores. In addition some of the columnwise structure is removed, while some still remains. The remaining structure is due to the $\|X_j\|_2$ component of the scores $\bar{S}_{ij} = \|\bar{W}_{ij}\| \|X_j\|_2$.

### 4.2 Comparisons with Other Compression Algorithms

The average computational cost of both NoWag-P and NoWag-VQ scales linearly with the size of the weight matrices, and thus the size of the LLM. Since $d_{\text{in}}$ and $d_{\text{out}}$ are of roughly the same magnitude in a modern LLM, NoWag-P and NoWag-VQ offers a significant speedup for compression over linear feedback based pruning methods, whose computational complexity scales cubically with $d_{\text{in}}$.

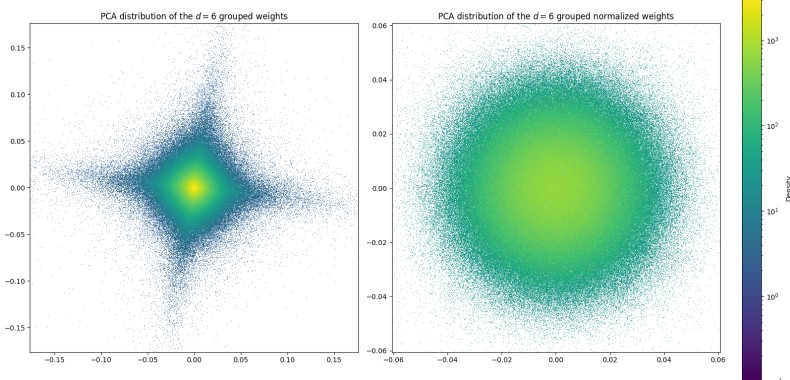

Figure 3: 2d PCA visualization of the distribution of $d = 6$ grouped entries from W and $\bar{\text{W}}$. Densities are plotted at log scale. Normalization reshapes the distribution into a more "ball-shaped distribution.

**Pruning:** Several parallels can be drawn between our approach and Wanda. First, without normalization, NoWag-P is equivalent to Wanda without output grouped pruning. Second, Wanda takes advantage of the same phenomenon described in Section 4.1 through output group pruning, only normalizing along $d_{out}$. However, output grouped pruning used by Wanda is not as strong as our normalization method in two aspects. First, normalization is only performed on the output dimensions; this breaks up the row wise structure, but some column-wise structure will still remain. Second, group-wise normalization cannot be used with semi-structured pruning or quantization. In comparison, our normalization method directly breaks up both row and column structures. Furthermore, our normalization method is applied directly to the, allowing for support for semi-structured pruning and other compression algorithms such as quantization.

**Quantization** Kmeans has been explored for LLM PTQ in several works. In many VQ algorithms, it is used to initialize before optimizing the quantization (Van Baalen et al., 2024; Liu et al., 2024; Egiazarian et al., 2024). For scalar quantization, SqueezeLLM (Kim et al., 2023) has employed weighted K-means using the diagonal of the fisher information as weights. Our algorithm has several key differences from those aforementioned. First, we use K-means *only*, without any computationally expensive optimization procedures required by previous VQ algorithms. Second, our weights are simply the second moment of the sample activations which can be calculated without a backwards pass.

## 5 Experiments

**Models**. We evaluate NoWag on two popular families of models Llama 2 (Llama-2 7B/13B/70B) (Touvron et al., 2023) and Llama-3 8B/70B (Grattafiori et al., 2024). Due to resource and time constraints, we did not apply NoWag-VQ to the Llama-3 70B model; therefore, for the Llama 3 family, we report quantization performance only for the Llama-3 8B model.

**Baselines** A particular focus in the research community is "one-shot" compression methods for large language models (LLMs). Here, "one-shot" refers to directly compressing the model based on the calibration data without fine-tuning to adjust the compressed model parameters. Such compression methods are desirable because they have minimal computational overhead associated with it. However in more recent works, for "extreme" compression, roughly < 3 bits per value, it has been demonstrated that specialized finetuning post quantization can significantly recover performance lost during compression (Malinovskii et al., 2024; Egiazarian et al., 2024; Tseng et al., 2024a).

The focus of our paper is *how to compress*. As a result, we primary focus on "one-shot" compression algorithms. The SOTA for one-shot VQ for LLMs at 2 bits per value is QuIP# (Tseng et al., 2024a), so we make comparisons against it. QuIP# incorporates VQ with Hammard incoherence matrices and an $E_8$ structured codebook. We did not compare against QTIP (Tseng et al., 2024b) as our focus was on VQ rounding methods, and because Trellis coding can be extended to any VQ rounding method (Tseng et al., 2024b).

For pruning, we compare NoWag-P against Wanda (Sun et al., 2023), an unstructured and semi-structured SOTA pruning algorithm. As discussed previously, the key difference between Wanda and NoWag-P is our normalization method. Wanda does no normalization, and only does output group wise pruning for unstructured pruning only, while for semi-structured pruning, Wanda uses just simple greedy pruning based on the element-wise importance scores. As such, a comparison between NoWag-P and Wanda serves to highlight the impact of our normalization scheme in both a unstructured and semi-structured pruning scheme.

**Calibration dataset** We use 128 samples at the model's native sequence length (4096 for the Llama 2 family and 8192 for the Llama 3 family) of the RedPajama 1T dataset (Weber et al., 2024) as our calibration data for both pruning and quantization. This is the same dataset used by QuIP#, which uses 6144 samples, or 48x times more data.

**Evaluation** To evaluate NoWag-VQ, we follow standard evaluation metrics for quantized models of measuring the perplexity on the test splits of the C4 (Dodge et al., 2021) and Wikitext2 (Merity et al., 2016). Furthermore, we performed task specific sequence classification zero shot accuracy through the Eleuther AI LM Harness (Gao et al., 2024). For equal comparison with QuIP#, we used the same version of LM Harness (Version 0.3.0). The exact tasks are listed in appendix B. It is worth noting that the zeroshot tasks have some intrinsic randomness, even FP16 numbers can disagree by up to 0.5%, due to their element of randomness.

## 5.1 Quantization Evaluation

| Method | Bits | Wiki2 ($\downarrow$) | C4 ($\downarrow$) |
|---|---|---|---|
| fp16 (2-7B) | 16 | 5.12 | 6.63 |
| fp16 (2-13B) | 16 | 4.57 | 6.05 |
| fp16 (2-70B) | 16 | 3.12 | 4.97 |
| fp16 (3-8B) | 16 | 5.54 | 7.01 |
| QUIP (2-7B) # | 2 | 8.23 | 10.8 |
| NoWag-VQ (2-7B) | 2.02 | **7.07** | **9.12** |
| QuIP # (2-13B) | 2 | 6.06 | 8.07 |
| NoWag-VQ (2-13B) | 2.01 | **5.93** | **7.94** |
| QuIP # (2-70B) | 2 | 4.16 | 6.01 |
| NoWag-VQ (2-70B) | 2.02 | **4.15** | **5.94** |
| QuIP# (3-8B) | 2 | 13.8 | 15.6 |
| NoWag-VQ (3-8B) | 2.02 | **10.68** | **11.92** |

Table 1: Perplexity on WikiText2 and C4 for 2-bit quantized Llama 2 (7B/13B/70B) and Llama 3 (8B) models without fine-tuning. Evaluations were performed at the models' native sequence lengths (4096 for Llama 2 and 8192 for Llama 3).

| Method | Bits | Wiki2 ($\downarrow$) | C4 ($\downarrow$) |
|---|---|---|---|
| fp16 (2-7B) | 16 | 5.12 | 6.63 |
| fp16 (2-13B) | 16 | 4.57 | 6.05 |
| fp16 (2-70B) | 16 | 3.12 | 4.97 |
| AQLM (2-7B) | 2.02 | 6.59 | 8.54 |
| NoWag-VQ (2-7B) | 2.02 | **6.51** | **8.50** |
| AQLM (2-13B) | 1.97 | 5.60 | 7.49 |
| NoWag-VQ (2-13B) | 2.01 | **5.53** | **7.39** |
| AQLM (2-70B) | 2.07 | **3.94** | **5.72** |
| NoWag-VQ (2-70B) | 2.02 | 3.99 | 5.77 |

Table 2: Perplexities for WikiText2 and C4 with blockwise finetuning for 2-bit Quantized Llama-2 7B/13B/70B compared with AQLM.

For Llama-2 7B/13B and Llama-3 8B, VQ was performed with groups of $d = 6$ elements together. This means that the codebook will be able to fit inside the L1 cache of an Nvidia A6000 GPU, enabling fast decoding. For Llama-2 70B, VQ was performed with groups of $d = 7$ elements together, since the relative overhead of the codebook was diminished

---

[1]QuIP# accuracies are taken from CALDERA (Saha et al., 2024).

|  | Bits | Wino (↑) | RTE (↑) | PiQA (↑) | ArcE (↑) | ArcC (↑) | Avg Acc (↑) |
|---|---|---|---|---|---|---|---|
| FP16 (2-7B) | 16 | 67.3 | 63.2 | 78.5 | 69.3 | 40.0 | 63.66 |
| FP16 (2-13B) | 16 | 69.5 | 61.7 | 78.8 | 73.2 | 45.6 | 65.76 |
| FP16 (2-70B) | 16 | 77.0 | 67.9 | 81.1 | 77.7 | 51.1 | 70.96 |
| FP16 (3-8B) | 16 | 73.5 | 68.6 | 79.7 | 80.1 | 50.2 | 70.42 |
| QuIP# (2-7B ) | 2 | 61.7 | **57.8** | 69.6 | 61.2 | 29.9 | 56.04 |
| NoWag-VQ (2-7B) | 2.02 | **64.4** | 54.5 | **73.6** | **60.7** | **31.7** | **56.99** |
| QuIP # (2-13B) | 2 | 63.6 | 54.5 | 74.2 | **68.7** | 36.2 | 59.44 |
| NoWag-VQ (2-13B) | 2.01 | **68.1** | **62.5** | **75.9** | 67.3 | **37.9** | **62.34** |
| QuIP # (2-70B) | 2 | 74.2 | **70.0** | 78.8 | **77.9** | **48.6** | **69.9** |
| NoWag-VQ (2-70B) | 2.02 | **74.5** | 69.0 | **79.4** | 75.4 | 46.2 | 68.9 |
| QuIP # (3-8B) | 2 | 63.2 | 52.7 | 67.6 | 57.6 | 28.2 | 53.86 |
| NoWag-VQ (3-8B) | 2.02 | **67.7** | **53.0** | **72.3** | **68.4** | **33.2** | **58.93** |

Table 3: Zeroshot sequence classification accuracies (%) across 5 tasks and the average accuracies of Quantized Models without finetuning. [1]

| Method | Bits | Wikitext2 PPL (↓) | | | C4 PPL (↓) | | |
|---|---|---|---|---|---|---|---|
| | | 2-7B | 2-13B | 2-70B | 2-7B | 2-13B | 2-70B |
| Dense | 0% | 5.47 | 4.88 | 3.32 | 6.97 | 6.47 | 5.52 |
| *One Shot* | | | | | | | |
| GPTVQ | 2.125 | 8.23 | 6.5 | 4.64 | - | - | - |
| ClusComp⁻ | < 2.01 | 52.38 | 22.9 | 9.84 | 50.08 | 24.47 | 13.96 |
| NoWag-VQ | < 2.02 | **7.59** | **6.37** | **4.41** | **9.28** | **8.16** | **6.34** |
| *Blockwise Finetuned* | | | | | | | |
| ClusComp | < 2.01 | 7.5 | 6.17 | 4.83 | 10.29 | 8.49 | 7.02 |
| NoWag-VQ | < 2.02 | **7.01** | **5.93** | **4.25** | **8.07** | **7.64** | **6.18** |

Table 4: Wikitext2 and C4 Perplexities of NoWag-VQ, GPTVQ, and ClusComp at ~ 2 bits per value for Llama-2 7B/13B/70B at 2048 context length. One shot and blockwise finetuned results are provided

with the larger model size. While larger, this codebook is still able to fit inside the L1 cache of a Nvidia H100 GPU. In Table 3 we compare the Zero Shot accuracies of NoWag-VQ against QuIP#. In Table 1 we compare the perplexities of NoWag-VQ against QuIP#, both at ~ 2 bits per value. NoWag-VQ outperforms QuIP# in perplexity for all models for both Wikitext2 and C4. In terms of zero shot accuracies, NoWag-VQ beats QuIP# for all models beyond Llama-2 70B. We would like to draw attention to how NoWag-VQ is able to significantly improve the quantized performance of Llama-3-8B compared with QuIP#. This is worth noting as the Llama-3 family of models is significantly harder to quantize (Huang et al., 2024).

We also examine the performance of NoWag-VQ beyond the "one-shot" compression regime. Existing literature has proposed several methods for post quantization finetuning. One popular method is finetuning the remaining continuous parameters of each transformer block to minimize the block output errors (Egiazarian et al., 2024). Another is model-wise finetuning to minimize the overall Kullback–Leibler divergence with the original model, optimizing over the continuous (Tseng et al., 2024a), and the discrete parameters (Malinovskii et al., 2024). Because of our limited computational resources, we were only able to perform block-wise finetuning. We compare the perplexities of those models against those of AQLM (Egiazarian et al., 2024) in table 2. NoWag-VQ outperforms AQLM for Llama 2 7B and 13B, but falls short for Llama-2 70B. We suspect this is due to AQLM using $d = 8$ VQ rather than $d = 7$, which allows for more than 4x the parameters. However, our codebook fits into the L1 cache of an H100 and AQLM's does not.

For additional baselines, we compared the performance of NoWag-VQ against two other recent LLM VQ methods, GPTVQ (Van Baalen et al., 2024), and ClusComp (Liao & Monz,

2025) in Table 4. ClusComp consists of several variants, including a zero-shot compression variant ClusComp$^-$, and a blockwise finetuned variant, ClusComp. We included our one-shot and blockwise finetuned variants to provided an apples to apples comparison. We believe these comparisons offer additional clarity for the benefits for the two main differences between NoWag and ClusComp. First, ClusComp does not incorporate any normalization, unlike our method. Second, ClusComp performs vanilla vector K-means, rather than weighted K-means, meaning that ClusComp has no data awareness and treats all input channels with equal importance. The benefits of these methods are especially evident in the one-shot regime.

## 5.2 Pruning Evaluation

| Method | Sparsity | Wikitext2 PPL ($\downarrow$) | | | | | C4 PPL ($\downarrow$) | | | | |
|---|---|---|---|---|---|---|---|---|---|---|---|
| | | 2-7B | 2-13B | 2-70B | 3-8B | 3-70B | 2-7B | 2-13B | 2-70B | 3-8B | 3-70B |
| Dense | 0% | 5.12 | 4.57 | 3.12 | 5.54 | 2.58 | 6.63 | 6.05 | 4.97 | 7.01 | 5.78 |
| Wanda | 50% | 6.46 | 5.58 | 3.97 | 9.06 | 5.34 | 8.39 | 7.47 | 5.77 | 10.19 | 7.00 |
| NoWag-P | 50% | **6.37** | **5.49** | **3.89** | **8.32** | **4.95** | **8.27** | **7.35** | **5.71** | **9.67** | **6.81** |
| Wanda | 4:8 | 8.07 | 6.55 | 4.49 | 13.39 | 6.50 | 10.19 | 8.68 | 6.39 | 13.95 | 7.95 |
| NoWag-P | 4:8 | **8.04** | **6.47** | **4.45** | **12.66** | **6.24** | **10.17** | **8.67** | **6.38** | **13.86** | **7.69** |
| Wanda | 2:4 | 11.35 | 8.36 | 5.20 | **22.42** | 8.29 | **13.80** | **10.96** | 7.19 | **21.63** | 9.63 |
| NoWag-P | 2:4 | **11.14** | **8.28** | **5.17** | 24.0 | **7.52** | 13.91 | 11.05 | **7.23** | 23.5 | **9.18** |

Table 5: Wikitext2 and C4 Perplexities NoWag-P and Wanda at 50% unstructured, and 4:8 and 2:4 semistructured pruning for Llama-2 7B/13B/70B and Llama-3 8B/70B. Context length was at the model's native context length, 4096 for Llama-2 and 8192 for Llama-3.

| Method | Sparsity | Avg Zero Shot Accuracy ($\uparrow$) | | | | |
|---|---|---|---|---|---|---|
| | | 2-7B | 2-13B | 2-70B | 3-8B | 3-70B |
| Dense | 0% | 63.66 | 65.76 | 70.96 | 70.42 | 75.89 |
| Wanda | 50% | 60.24 | **63.66** | 70.16 | **63.49** | **73.45** |
| NoWag-P | 50% | **60.48** | 63.57 | **70.28** | 62.93 | 72.3 |
| Wanda | 4:8 | **58.27** | **61.32** | **68.7** | **57.84** | 70.42 |
| NoWag-P | 4:8 | 56.71 | 60.6 | 68.43 | 57.46 | **70.76** |
| Wanda | 2:4 | **55.37** | **58.24** | 66.73 | **52.59** | **68.08** |
| NoWag-P | 2:4 | 54.3 | 58.14 | **66.95** | 51.21 | 67.71 |

Table 6: Zeroshot Accuracies for Llama-2 7B/13B/70B and Llama-3 8B/70B pruned using NoWag-P and Wanda for 50% unstructured, 2:4 semi-structured, and 4:8 semi-structured

Because Wanda did not report C4 perplexities, we modified the code to compute them, furthermore we added support for pruning the Llama-3 family of models. Because of this, several libraries had to be upgraded from what the original Wanda paper used, resulting perplexities in Wikitext2 that are slightly different to those reported in Wanda.

Table 5 shows a comparison of the language modeling abilities of NoWag-P and Wanda pruned models, measured through Wikitext2 and C4 perplexity. Three different levels of pruning patterns were evaluated, 50% unstructured, 4:8 semi-structured, and 2:4 semi-structured pruning. NoWag-P uniformly outperforms Wanda at 50% and 4:8 semi-structured pruning. This empirically demonstrates the benefits of the NoWag normalizer. However at 2:4 semi-structured pruning, NoWag-P only roughly matches the performance of Wanda. We believe that this is due to the more structured pattern, which negates the impact of the normalizer. To confirm, we conducted a sweep of pruning using NoWag-P and Wanda for a range of N:M structures for Llama-2-13B and Llama-3-8B, the results are visualized in Figure 4. The relative reduction of C4 perplexity between NoWag-P and rapidly diminishes as the pattern becomes more structured ($N$ becomes smaller).

Table 6 shows a comparison of the average zero shot task accuracy for NoWag-P and Wanda pruned models. Once again, three different levels of pruning patterns were eval-

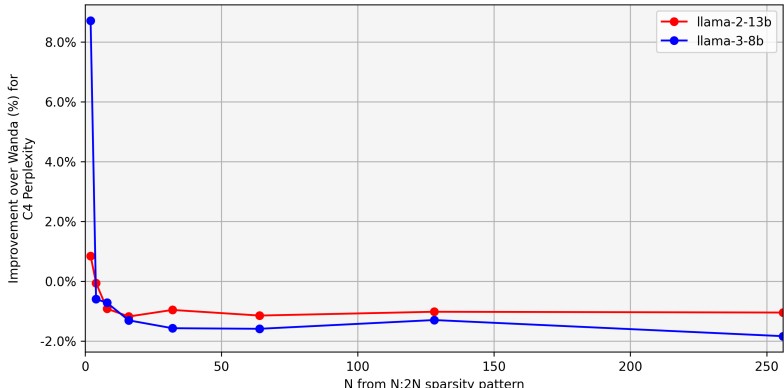

Figure 4: Relative difference in C4 perplexity NoWag-P between Wanda: (NoWag Perplexity)/(Wanda Perplexity) − 1. Calculated for a range of semi structured patterns for Llama-2-13B and Llama-3-8B

uated, 50% unstructured, 4:8 semi-structured, and 2:4 semi-structured pruning. For most models across most pruning methods NoWag-P and Wanda produce models similar average zero shot task accuracy.

## 6 Conclusion

In this work, we introduced NoWag, a novel framework unifying pruning and quantization under a common normalization-based approach. Our experimental results demonstrate that NoWag-P improves upon existing pruning techniques in maintaining language modeling accuracy, while NoWag-VQ achieves superior quantization performance with substantially less calibration data. By leveraging a structured normalization strategy, NoWag reduces the sensitivity of compression to outlier weights and enhances the efficiency of both pruning and quantization. These findings establish NoWag as a scalable and adaptable compression paradigm for LLMs, facilitating their deployment in real-world applications with reduced computational costs.

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

# Appendix

# A    Detailed Formulation of Quantization

To quantize a model, we map each weight entry or a contiguous vector of weight entries to a codebook. Without loss of generality, we assume that use a vector quantization algorithm where we quantize every $d$ parameters, $w_{i,j:j+d}$ together. Then quantization results in the following:

- A **codebook** $\mathcal{C} = \{c_1...c_k | c_k \in \mathbb{R}^d\}$
- A **mapping** $\mathcal{M}(w_{i,j:j+d}) = c_l \in \mathcal{C}$.

when we represent the quantized weights for inference, these mappings become a string of bits of length $\lceil |\mathcal{C}| \rceil$. Therefore the resulting in a bits per value of $(\log_2(\lceil |\mathcal{C}| \rceil) + \epsilon)\frac{1}{d}$, where $\epsilon$ is the bits needed to represent the overhead of normalization parameters, codebooks, etc. Note that we can inverse this relationship to find that if we have a target bits per values $\sim n_{bpv}$, the size of the codebook should be $|\mathcal{C}| = 2^{n_{bpv}d}$. The main benefit of vector quantization is that it allows for the quantization codebook to be better shaped to the weights. However, the size of the codebook increases exponentially with the dimension $d$, which leads to $\epsilon/d$ no longer becoming a negligible quantity compared with the bits used to encode each value. Furthermore, for fast inference, $\mathcal{C}$ must fit inside the L1 cache of a GPU. This has lead to a line of work on more efficient encoding schemes, such as trellis encoding schemes pioneered by (Tseng et al., 2024b).

## A.1    Weighted Vector K-Means Formulation

As discussed in section 4, for quantization we used weighted vector K-Means, this consists of two steps, an assignment step and an update step, below we explicitly write each step. Traditionally, K-Means is run with several random initializations, and allowed to converge for each initialization. However, in the interests of runtime, for each layer, we only initialized once using the K-means++ algorithm (Arthur & Vassilvitskii, 2007), and only performed 100 assignment update step pairs. We did observe increasing performance scaling from 20 to 100 assignment steps, therefore we believe that the results reported in tables 1 3 do not demonstrate the full performance of NoWag-VQ.

**Assignment step:** For each vector $\bar{W}_{i,j:j+d}$ we select the mappings to such that the weighted l2 norm is minimized.

$$\mathcal{M}(\bar{W}_{i,j:j+d}) = \underset{c_l \in \mathcal{C}}{\arg\min} \left(\bar{W}_{i,j:j+d} - c_l\right)^T \left(\text{diag}(XX^T)_{j:j+d} \odot \left(\bar{W}_{i,j:j+d} - c_l\right)\right) \quad (4)$$

**Update step:** For each vector in the codebook, we take the weighted average of the assignments:

$$c_l = \left(\sum_{i,j \forall \mathcal{M}(\bar{W}_{i,j:j+d})=c_l} \text{diag}(XX^T)_{j:j+d} \odot \bar{W}_{i,j:j+d}\right) \oslash \left(\sum_{i,j \forall \mathcal{M}(\bar{W}_{i,j:j+d})=c_l} \text{diag}(XX^T)_{j:j+d}\right) \quad (5)$$

# B    Zero Shot Task Descriptions

NoWag-P was evaluated on zero-shot accuracy as noted Tables 10, 11, and 12. The classification sequence classification tasks are as follows:

1. **WinoGrande** (Sakaguchi et al., 2021) — WINOGRANDE is a large-scale dataset of 44k problems based on the Winograd Schema Challenge, specifically crafted to minimize biases in training data. It features a two-choice fill-in-the-blank format that requires deep commonsense reasoning.

2. **RTE** (Dagan et al., 2006; Giampiccolo et al., 2007; Dzikovska et al., 2013) — The Recognizing Textual Entailment (RTE) Challenge has participants determine semantic relationships like entailment, contradiction, or neutrality between sentence pairs.

3. **PIQA** (Bisk et al., 2020) — PIQA is a benchmark dataset for physical commonsense reasoning, having AI answer questions about everyday interactions without direct physical experience.

4. **ARC-e** (Clark et al., 2018) — A subset of the AI2 Reasoning Challenge (ARC), ArcE consists of multiple-choice questions designed to assess grade-school level knowledge and represents the "Easy" portion of the dataset.

5. **ARC-c** (Clark et al., 2018) — The ARC-Challenge subset follows the same format as ARC-Easy but includes only questions that baseline algorithms previously failed to answer correctly.

## C  Quantization Additional Evaluations

We include additional comparisons with three more VQ algorithms, AQLM (Egiazarian et al., 2024), VPTQ (Liu et al., 2024) and CALDERA (Saha et al., 2024). AQLM (Additive Quantiztation for Language Models) performs quantization through additive multi code-book VQ. VPTQ combindeds are vector quantization extension of GPTQ (Frantar et al., 2022) with residual quantization. CALDERA builds ontop of QuIP# by adding additional quantized low rank matrices, and as a result, requires higher bits per value.

For AQLM and VPTQ, their appendices included zero shot ablation results for perplexity of Wikitext2 and C4. For VPTQ, because a very detailed ablation was chosen, we simply chose the best performing zero-shot quantization. For CALDERA, we chose the compression with the least equivalent bits per value. Perplexities are shown in table 7, and zero shot results with CALDERA are show in table 8. We can see that our algorithm performs competitively to these modern VQ algorithms as well.

| Method | Bits | Wiki2 ($\downarrow$) | C4 ($\downarrow$) |
|---|---|---|---|
| fp16 (2-7B) | 16 | 5.12 | 6.63 |
| fp16 (2-13B) | 16 | 4.57 | 6.05 |
| fp16 (3-8B) | 16 | 5.54 | 7.01 |
| fp16 (2-70B) | 16 | 3.12 | 4.97 |
| AQLM (2-7B) | 2.02 | 8.18 | 10.59 |
| QUIP # (2-7B) | 2 | 8.23 | 10.8 |
| CALDERA (7B) | 2.1 | 7.37 | 9.74 |
| NoWag-VQ (2-7B) | 2.02 | **7.07** | **9.02** |
| QuIP # (2-13B) | 2 | 6.06 | 8.07 |
| CALDERA (2-13B) | 2.08 | 6.04 | 7.98 |
| VPTQ (2-13B) | 2.07 | 6.02 | 7.96 |
| NoWag-VQ (2-13B) | 2.01 | **5.93** | **7.94** |
| QuIP# (3-8B) | 2 | 13.8 | 15.6 |
| CALDERA | 2.1 | **10.6** | **11.8** |
| NoWag-VQ (3-8B) | 2.02 | 10.68 | 11.92 |
| QuIP# (2-70B) | 2 | 4.16 | 6.01 |
| CALDERA (2-70B) | 2.1 | **4.11** | 5.95 |
| NoWag-VQ (2-70B) | 2.01 | 4.15 | **5.94** |

Table 7: Performance comparison of different methods on Wiki2 and C4 datasets.

| | Bits | Wino (↑) | RTE (↑) | PiQA (↑) | ArcE (↑) | ArcC (↑) | Avg Acc (↑) |
|---|---|---|---|---|---|---|---|
| FP16 (2-7B) | 16 | 67.3 | 63.2 | 78.5 | 69.3 | 40.0 | 63.66 |
| FP16 (2-13B) | 16 | 69.5 | 61.7 | 78.8 | 73.2 | 45.6 | 65.76 |
| FP16 (3-8B) | 16 | 73.5 | 68.6 | 79.7 | 80.1 | 50.2 | 70.42 |
| FP16 (2-70B) | 16 | 77.0 | 67.9 | 81.1 | 77.7 | 51.1 | 70.96 |
| QuIP# (2-7B ) | 2 | 61.7 | **57.8** | 69.6 | 61.2 | 29.9 | 56.04 |
| Caldera (2-7B) | 2.1 | 63.7 | **62.1** | 72.3 | **60.9** | **31.7** | **58.14** |
| NoWag-VQ (2-7B) | 2.02 | **64.4** | 54.5 | **73.6** | 60.7 | **31.7** | 56.99 |
| QuIP # (2-13B) | 2 | 63.6 | 54.5 | 74.2 | 68.7 | 36.2 | 59.44 |
| Caldera (2-13B) | 2.08 | 66.9 | 61.0 | **76.0** | **69.5** | 37.2 | 62.12 |
| NoWag-VQ (2-13B) | 2.01 | **68.1** | **62.5** | 75.9 | 67.3 | **37.9** | **62.34** |
| QuIP # (3-8B) | 2 | 63.2 | 52.7 | 67.6 | 57.6 | 28.2 | 53.86 |
| Caldera (3-8B) | 2.1 | 66.9 | **58.5** | 71.8 | 68.2 | **34.3** | **59.94** |
| NoWag-VQ (3-8B) | 2.02 | **67.7** | 53.0 | **72.3** | **68.4** | 33.2 | 58.93 |
| QuIP # (2-70B) | 2 | 74.2 | **70.0** | 78.8 | 77.9 | **48.6** | **69.9** |
| Caldera (2-70B) | 2.1 | **75.5** | 69.3 | **79.8** | **76.9** | 47.7 | 69.84 |
| NoWag-VQ (2-70B) | 2.02 | 74.5 | 69.0 | 79.4 | 75.4 | 46.2 | 68.9 |

Table 8: Zeroshot accuracies (%) across 5 tasks and the average accuracies of Quantized Models without finetuning.

## D   Additional Pruning Results

| | | Wikitext2 PPL (↓) | | | | |
|---|---|---|---|---|---|---|
| | Sparsity | 2-7b | 2-13b | 2-70b | 3-8b | 3-70b |
| SparseGPT | 50% | 6.51 | 5.63 | 3.98 | 8.53 | 5.31 |
| NoWag-P | 50% | **6.37** | **5.49** | **3.89** | **8.32** | **4.95** |
| SparseGPT | 4:8 | **7.99** | 6.58 | 4.59 | **10.92** | 6.56 |
| NoWag-P | 4:8 | 8.04 | **6.47** | **4.45** | 12.66 | **6.24** |
| SparseGPT | 2:4 | **10.23** | 8.29 | 5.38 | **14.31** | 8.62 |
| NoWag-P | 2:4 | 11.14 | **8.28** | **5.17** | 24.0 | **7.52** |

Table 9: Comparison of NoWag-P and SparseGPT (Frantar & Alistarh, 2023) perplexities for Llama-2 7B/13B/70B and Llama-3 8B/70B

| | Sparsity | Wino (↑) | RTE (↑) | PiQA (↑) | ArcE (↑) | ArcC (↑) | Avg Acc (↑) |
|---|---|---|---|---|---|---|---|
| FP16 (2-7B) | 0% | 67.3 | 63.2 | 78.5 | 69.3 | 40.0 | 63.66 |
| FP16 (2-13B) | 0% | 69.5 | 61.7 | 78.8 | 73.2 | 45.6 | 65.76 |
| FP16 (2-70B) | 0% | 77.0 | 67.9 | 81.1 | 77.7 | 51.1 | 70.96 |
| FP16 (3-8B) | 0% | 73.5 | 68.6 | 79.7 | 80.1 | 50.2 | 70.42 |
| FP16 (3-70B) | 0% | 80.7 | 69.0 | 82.5 | 86.8 | 60.4 | 75.89 |
| Wanda (2-7B) | 50% | **66.9** | 55.6 | 75.6 | **66.2** | 37.0 | 60.24 |
| NoWag-P (2-7B) | 50% | 65.7 | **60.7** | 75.7 | 65.5 | 34.9 | **60.48** |
| Wanda (2-13B) | 50% | 68.9 | 58.5 | **78.4** | **71.6** | 41.0 | 63.66 |
| NoWag-P (2-13B) | 50% | 68.9 | 59.6 | 77.8 | 71.3 | 41.0 | 63.57 |
| Wanda (2-70B) | 50% | **76.9** | 69.3 | 80.5 | **75.9** | 48.2 | 70.16 |
| NoWag-P (2-70B) | 50% | 76.6 | **71.1** | 80.7 | 75.5 | 47.4 | **70.28** |
| Wanda (3-8B) | 50% | **71.0** | **59.9** | 74.9 | 71.4 | 40.3 | **63.49** |
| NoWag-P (3-8B) | 50% | 70.0 | 56.7 | **75.8** | 71.7 | 40.4 | 62.93 |
| Wanda (3-70B) | 50% | **78.0** | **70.0** | 81.3 | 83.0 | 55.0 | **73.5** |
| NoWag-P (3-70B) | 50% | 76.7 | 67.9 | 81.2 | 82.8 | 52.9 | 72.30 |

Table 10: Zeroshot accuracies for each task for 50% Pruning

| | Sparsity | Wino (↑) | RTE (↑) | PiQA (↑) | ArcE (↑) | ArcC (↑) | Avg Acc (↑) |
|---|---|---|---|---|---|---|---|
| FP16 (2-7B) | 0% | 67.3 | 63.2 | 78.5 | 69.3 | 40.0 | 63.66 |
| FP16 (2-13B) | 0% | 69.5 | 61.7 | 78.8 | 73.2 | 45.6 | 65.76 |
| FP16 (2-70B) | 0% | 77.0 | 67.9 | 81.1 | 77.7 | 51.1 | 70.96 |
| FP16 (3-8B) | 0% | 73.5 | 68.6 | 79.7 | 80.1 | 50.2 | 70.42 |
| FP16 (3-70B) | 0% | 80.7 | 69.0 | 82.5 | 86.8 | 60.4 | 75.89 |
| Wanda (2-7B) | 4:8 | **65.35** | **58.12** | **73.61** | **62.46** | **31.83** | **58.27** |
| NoWag-P (2-7B) | 4:8 | 64.7 | 54.2 | 72.7 | 61.7 | 30.2 | 56.71 |
| Wanda (2-13B) | 4:8 | 68.98 | 55.96 | **75.79** | **67.47** | **38.4** | **61.32** |
| NoWag-P (2-13B) | 4:8 | **69.0** | **57.0** | 75.0 | 65.5 | 36.5 | 60.60 |
| Wanda (2-70B) | 4:8 | 74.9 | **67.87** | **79.54** | **74.71** | **46.5** | **68.7** |
| NoWag-P (2-70B) | 4:8 | **75.6** | 67.2 | 79.3 | 74.0 | 46.2 | 68.43 |
| Wanda (3-8B) | 4:8 | **66.69** | 53.07 | **71.0** | **64.31** | **34.13** | **57.84** |
| NoWag-P (3-8B) | 4:8 | 65.0 | **54.5** | 70.7 | 63.6 | 33.4 | 57.46 |
| Wanda (3-70B) | 4:8 | 73.8 | **66.06** | **80.09** | 80.89 | **51.28** | 70.42 |
| NoWag-P (3-70B) | 4:8 | **76.1** | 65.7 | 79.7 | **81.4** | 50.9 | **70.76** |

Table 11: Zeroshot accuracies for each task for 4:8 Pruning

| | Sparsity | Wino (↑) | RTE (↑) | PiQA (↑) | ArcE (↑) | ArcC (↑) | Avg Acc (↑) |
|---|---|---|---|---|---|---|---|
| FP16 (2-7B) | 0% | 67.3 | 63.2 | 78.5 | 69.3 | 40.0 | 63.66 |
| FP16 (2-13B) | 0% | 69.5 | 61.7 | 78.8 | 73.2 | 45.6 | 65.76 |
| FP16 (2-70B) | 0% | 77.0 | 67.9 | 81.1 | 77.7 | 51.1 | 70.96 |
| FP16 (3-8B) | 0% | 73.5 | 68.6 | 79.7 | 80.1 | 50.2 | 70.42 |
| FP16 (3-70B) | 0% | 80.7 | 69.0 | 82.5 | 86.8 | 60.4 | 75.89 |
| Wanda (2-7B) | 2:4 | **60.5** | **58.5** | **70.1** | **57.6** | **30.2** | **55.37** |
| NoWag-P (2-7B) | 2:4 | **60.5** | 58.1 | 69.3 | 55.8 | 27.9 | 54.30 |
| Wanda (2-13B) | 2:4 | **65.8** | 54.5 | **73.1** | **63.8** | 34.0 | **58.2** |
| NoWag-P (2-13B) | 2:4 | 65.6 | **58.1** | 72.4 | 62.9 | 32.7 | 58.14 |
| Wanda (2-70B) | 2:4 | 73.6 | 64.6 | **78.9** | **72.9** | **43.2** | 66.70 |
| NoWag-P (2-70B) | 2:4 | **75.1** | **66.8** | 77.6 | 72.5 | 42.7 | **66.95** |
| Wanda (3-8B) | 2:4 | **59.9** | 52.7 | **67.5** | **56.9** | **25.9** | **52.59** |
| NoWag-P (3-8B) | 2:4 | 58.1 | 52.7 | 66.6 | 54.4 | 24.2 | 51.21 |
| Wanda (3-70B) | 2:4 | 71.7 | **63.9** | 78.1 | **78.5** | **48.2** | **68.08** |
| NoWag-P (3-70B) | 2:4 | **72.9** | 62.5 | **78.5** | 77.8 | 46.9 | 67.71 |

Table 12: Zeroshot accuracies for each task for 2:4 Pruning

# E  Inference Speedup

A primary objective of model compression is to reduce the memory footprint during inference. NoWag-P, at 50% sparsity, achieves a 2× reduction in memory required to store weights, while NoWag-VQ compresses weights to 2 bits per value—resulting in an 8× reduction compared to fp16 precision.

In addition to memory savings, our method also leads to inference speedups due to the alleviation of memory bandwidth bottlenecks and the use of structured sparsity. To illustrate this, in Table 13 we report the matrix multiplication runtimes for 2:4 NoWag-P on an NVIDIA A6000 GPU, matching the evaluation setup used in prior work such as SparseGPT, Wanda, and AQLM. Specifically, we benchmark matrix-vector multiplication on the gate projection layers of LLaMA-2 models of varying sizes (7B, 13B, and 70B).

|  | 2-7B | 2-13B | 2-70B |
|---|---|---|---|
| FP-16 | 0.137ms | 0.205ms | 0.673ms |
| NoWag-P | 0.117ms (1.173x) | 0.15ms (1.373x) | 0.401ms (1.679x) |

Table 13: Benchmarked matrix-vector multiplication on the gate projection layers of LLaMA-2 models of varying sizes (7B, 13B, and 70B), compared between the original dense FP-16 matrices and the NoWag-P compressed matrices

# F   Additional Algorithms

---

**Algorithm 4** VQ Padding

---

**Require:** Normalized weight matrix $\bar{W} \in \mathbb{R}^{d_\text{out} \times d_\text{in}}$, input dimension $d_\text{in}$, subvector dimension $d$
**Ensure:** Padded weight matrix $\bar{W}_\text{padded}$, padded input dimension $d_\text{in}^\text{padded}$
1: **if** $d_\text{in} \bmod d \neq 0$ **then**
2:   $\text{pad\_size} \leftarrow d - (d_\text{in} \bmod d)$
3:   $\text{pad\_value} \leftarrow \text{mean}(\bar{W})$
4:   Pad $\bar{W}$ with pad\_value to shape $(d_\text{out}, d_\text{in} + \text{pad\_size})$
5:   $d_\text{in}^\text{padded} \leftarrow d_\text{in} + \text{pad\_size}$
6: **else**
7:   $\bar{W}_\text{padded} \leftarrow \bar{W}$
8:   $d_\text{in}^\text{padded} \leftarrow d_\text{in}$
9: **end if**
10: **return** $\bar{W}_\text{padded}, d_\text{in}^\text{padded}$

---

**Algorithm 5** NoWag Pruning (NoWag-P)

---

**Require:** Weight matrix $W \in \mathbb{R}^{d_\text{out} \times d_\text{in}}$, activation samples $X \in \mathbb{R}^{m \times d_\text{in}}$, target sparsity $s$, sparsity pattern $P \in \{\text{unstructured}, \text{N:M}\}$
**Ensure:** Pruned weight matrix $\hat{W}$
1: $(r^{(1)}, r^{(2)}, \bar{W}) \leftarrow \texttt{NoWag\_Normalization}(W)$ {Apply normalization}
2: $S_{ij} \leftarrow \bar{W}_{ij}^2 \cdot \|X_j\|_2^2, \forall i \in [1, d_\text{out}], \forall j \in [1, d_\text{in}]$ {Importance scores}
3: **if** Pattern is unstructured **then**
4:   $\tau \leftarrow \{s\text{-th percentile of all elements in } S\}$
5:   $M \leftarrow \mathbf{1}_{S > \tau}$ {Binary mask where $M_{ij} = 1$ if $S_{ij} > \tau$, else 0}
6: **else if** Pattern is N:M **then**
7:   **for** each row $i \in [1, d_\text{out}]$ **do**
8:     **for** $g = 0$ to $\lfloor d_\text{in}/M \rfloor - 1$ **do**
9:       $I_g \leftarrow \{gM + 1, gM + 2, \ldots, gM + M\}$ {Indices of group $g$}
10:      $T_g \leftarrow \text{top-N}(\{S_{ij} : j \in I_g\})$ {Indices of top N scores}
11:      $M_{ij} \leftarrow \begin{cases} 1 & \text{if } j \in T_g \\ 0 & \text{otherwise} \end{cases}, \forall j \in I_g$ {Set mask}
12:    **end for**
13:  **end for**
14: **end if**
15: $\hat{W} \leftarrow W \odot M$ {Element-wise multiplication}
16: **return** $\hat{W}$

---

---

**Algorithm 6** Weighted K-Means with K-Means++ Initialization

---

**Require:** Subvectors $\bar{W}_{\text{sub}} \in \mathbb{R}^{N \times d}$, importance weights $H_{\text{sub}} \in \mathbb{R}^{N \times d}$, number of centroids $n_{\text{centroids}}$, number of iterations $n_{\text{iter}}$
**Ensure:** Codebook $C \in \mathbb{R}^{n_{\text{centroids}} \times d}$, assignments $A \in \mathbb{R}^N$

1: **Initialize codebook** $C \in \mathbb{R}^{n_{\text{centroids}} \times d}$:
2: Select random indices $I = \{i_1, i_2, \ldots, i_{n_{\text{centroids}}}\}$ from $[1, N]$ without replacement
3: $C_l \leftarrow \bar{W}_{\text{sub}, i_l}, \forall l \in [1, n_{\text{centroids}}]$
4: **for** $t = 1$ to $n_{\text{iter}}$ **do**
5:     **Assignment step:**
6:     **for** each subvector $i \in [1, N]$ **do**
7:         $D_{i,l} \leftarrow \sum_{j=1}^{d} H_{\text{sub},i,j} \cdot (\bar{W}_{\text{sub},i,j} - C_{l,j})^2, \forall l \in [1, n_{\text{centroids}}]$
8:         $A_i \leftarrow \arg\min_l D_{i,l}$
9:     **end for**
10:    **Update step:**
11:    **for** each centroid $l \in [1, n_{\text{centroids}}]$ **do**
12:       $S_l \leftarrow \{i : A_i = l\}$
13:       **if** $S_l \neq \varnothing$ **then**
14:         **for** dimension $j \in [1, d]$ **do**
15:           $C_{l,j} \leftarrow \frac{\sum_{i \in S_l} H_{\text{sub},i,j} \cdot \bar{W}_{\text{sub},i,j}}{\sum_{i \in S_l} H_{\text{sub},i,j}}$ {Weighted average}
16:         **end for**
17:       **end if**
18:    **end for**
19:    **if** assignments haven't changed since last iteration **then**
20:       **break**
21:    **end if**
22: **end for**
23: **return** $C, A$

---

## G  Implementation Details

### G.1  Finetuning

Finetuning was performed in a blockwise fashion. For each transformer block, our objective was minimizing the l2 norm between the outputs of the original block and those of the quantized blocks. The parameters to optimize over were the codebooks, and the normalization vectors of each quantized layers, and the RMS norm parameters. In addition we initialized a bias vector, set to all zeros, for each linear layer in the block.

For Llama-2 7B/13B, finetuning was done using 128 samples of Red Pajamas (Weber et al., 2024), with 32 held out as a validation set. Optimization was done through Adam Kingma & Ba (2017), without any weight decay and a learning rate of $10^{-4}$. For Llama-2 70B, 256 samples of Red Pajamas was used with a learning rate $10^{-6}$. Configuration management for both one-shot and finetuning was done through hydra (Yadan, 2019).

## H  Ablations

### H.1  K-means iterations for NoWag-VQ

| | Wikitext2 PPL ($\downarrow$) | | | C4 PPL ($\downarrow$) | | |
|---|---|---|---|---|---|---|
| | (2-7B) | (2-13B) | (3-8B) | (2-7B) | (2-13B) | (3-8B) |
| QuIP# | 8.23 | 6.06 | 13.80 | 10.80 | 8.07 | 15.60 |
| NoWag-VQ $T = 1$ | **7.32** | 6.20 | **12.44** | **9.58** | 8.36 | **13.96** |
| NoWag-VQ $T = 10$ | 7.04 | **6.00** | 11.08 | 9.19 | **8.03** | 12.35 |
| NoWag-VQ $T = 40$ | 7.10 | 5.99 | 10.86 | 9.15 | 7.98 | 12.07 |
| NoWag-VQ $T = 100$ | 7.07 | 5.93 | 10.68 | 9.12 | 7.94 | 11.92 |

Table 14: Ablation for $T = 1, 10, 40$ against the default $T = 100$ for NoWag-VQ for Llama-2-7b/13b and Llama-3 8B at 2 bits per value. We listed the Wikitext2 and C4 perplexities below, **bolding** the smallest for which NoWag outperforms QuIP#. For Llama-2 7b and Llama-3 8B NoWag with only one iteration of K-means was able to outperform QuIP#, for Llama-2 13B, after 10 iterations, NoWag-VQ was able to outperform QuIP#. We also note that the performance of the model continues to increase as we increase the number of iterations.

### H.2  Row and Column Normalization

Across all models and all compression methods, normalizing along row and column outperforms normalizing along row or along column alone.

| | Wikitext2 PPL ($\downarrow$) | | C4 PPL ($\downarrow$) | |
|---|---|---|---|---|
| | (2-7b) | (3-8B) | (2-7b) | (3-8B) |
| NoWag-VQ col only | 7.45 | 11.45 | 9.54 | 12.53 |
| NoWag-VQ row only | 7.68 | 11.29 | 10.10 | 12.53 |
| NoWag-VQ row and column | **7.07** | **10.68** | **9.12** | **11.92** |

Table 15: Ablation for NoWag-VQ at 2 bits per value with row or col only normalization against the default of row and column normalization for Llama-2 7B and Llama-3 8B

| | Wikitext2 PPL (↓) | | | C4 PPL (↓) | | |
|---|---|---|---|---|---|---|
| | (2-7b) | (2-13B) | (3-8B) | (2-7b) | (2-13B) | (3-8B) |
| NoWag-P col only | 7.01 | 5.53 | 9.26 | 9.22 | 7.44 | 10.60 |
| NoWag-P row only | 6.42 | 5.82 | 8.49 | 8.37 | 7.88 | 9.78 |
| NoWag-P row and column | **6.37** | **5.49** | **8.32** | **9.12** | **7.35** | **9.67** |

Table 16: Ablation for NoWag-P at 50% unstructured sparsity with row or col only normalization against the default of row and column normalization for Llama-2 7B/13B and Llama-3 8B

## H.3 Calibration Dataset

| | Wikitext2 PPL (↓) | | C4 PPL (↓) | |
|---|---|---|---|---|
| Calibration dataset | (2-7b) | (3-8B) | (2-7b) | (3-8B) |
| Wikitext2 | **6.9** | 10.9 | 9.16 | 12.55 |
| C4 | 7.19 | 11.22 | 9.12 | 12.055 |
| RedPajamas (default) | 7.07 | **10.68** | **9.12** | **11.92** |

Table 17: Ablation for different calibration datasets, training splits of Wikitext2 and C4, compared with the default of RedPajamas, conducted on NoWag-VQ for Llama-2 7B and Llama-3 8B. 128 samples were used for each dataset.

