# OpenReview forum: "NoWag: A Unified Framework for Shape Preserving Com- pression of Large Language Models"
_colmweb.org/COLM/2025/Conference — COLM 2025_

### Official Review · Reviewer_H7VY · 2025-04-19

**Rating:** 6
**Confidence:** 4
**Ethics Flag:** 1

**Summary:**

The paper introduces NoWag, a framework combining vector quantization (NoWag-VQ) and pruning (NoWag-P) to compress large language models (LLMs). It achieves better performance than existing methods, reducing computational and memory requirements while maintaining accuracy.

**Questions To Authors:**

N/A

**Reasons To Accept:**

1. This framework achieves good performance.

2. This paper is well-written.

**Reasons To Reject:**

1. There's no inference speedup evaluation.

2. The technique contribution is insufficient, as I believe the only contribution is that normalization can help gain good performance for VQ and pruning.

3. Lack of ablation comparisons of proposed methods w/ and w/o column(row) normalization.

4. Comparisons with more advanced pruning approaches could help validate its SOTA performance, e.g., [SparseGPT](https://github.com/IST-DASLab/sparsegpt), [https://arxiv.org/abs/2409.17372](https://arxiv.org/abs/2409.17372).

---

> ### Author Response · Authors · 2025-06-02
>
> We thank Reviewer H7VY for their constructive feedback and for acknowledging the clarity of our writing and the performance benefits of our approach. Below we address the concerns raised in detail, due to the number of reasons raised, we split our response into several comments due to the character limit
>
> **Reason to Reject 1: There's no inference speedup evaluation**
>
> We would like to emphasize that a central goal of model compression is reducing the memory needed for inference. NoWag-P at 50% sparsity results in a corresponding halving of memory needed to store the weights, and NoWag-VQ at 2 bits per value reduces the memory needed by 8x compared to fp16.
>
> Nevertheless, we still observe inference speedups from the removal of memory bottlenecks and sparse matrices. We have included below an evaluation of the speedup of 2:4 NoWag-P as an example. We measured the wall clock of the matrix multiplication times on an A6000, which was the same setup as sparseGPT and Wanda and AQLM for inference speedup evaluation. Below we have included the benchmarks for matrix vector multiplication on a standard layer (gate projection) of the Llama-2 7B/13B/70B models.
>
> |         | 2-7B             | 2-13B           | 2-70B            |
> |---------|------------------|-----------------|------------------|
> | **FP-16**   |  0.137ms         | 0.205ms         | 0.673ms          |
> | **NoWag-P** | 0.117ms (1.173x) | 0.15ms (1.373x) | 0.401ms (1.679x) |
>
> **Reason to Reject 2: The technique contribution is insufficient, as I believe the only contribution is that normalization can help gain good performance for VQ and pruning.**
>
> We respectfully disagree that our only contribution is showing that normalization helps. To put it succinctly, we propose lightweight normalization and a data aware proxy loss. This proxy loss is elementwise decomposable, allowing for a local/global optimal solution to be found quickly.
>
> For VQ, this provides additional benefits; existing SOTA VQ methods rely on computationally expensive Hessian-based proxy losses that require iterative optimization and are sensitive to calibration data. In contrast, NoWag-VQ eliminates these dependencies entirely through normalization, achieving comparable performance in blockwise finetuning and superior performance in the challenging one-shot regime where Hessian-based methods struggle most. This represents a significant methodological advance: we achieve SOTA results while avoiding the core computational and stability issues that plague current approaches.

---

> > ### Author Response · Authors · 2025-06-02
> >
> > **Reason to Reject 3: Lack of ablation comparisons of proposed methods w/ and w/o column(row) normalization.**
> >
> > We conducted an ablation for both NoWag-VQ and NoWag-P (50% unstructured) for w and w/o row and column normalization. We conducted this ablation study on the smaller models, Llama 2-7B and Llama 3 8B for NoWag-VQ and Llama 2-7B/13B and Llama 3-8B for NoWag-P. We reported the Wikitext2 (Wiki2) and Across all models and all compression methods, normalizing along row and column outperforms normalizing along row or along column.
> >
> > **NoWag-VQ ablation**
> > |                         | Wiki2 (2-7b) (↓) | Wiki2 (3-8B) (↓)  | C4 (2-7b) (↓)  | C4 (3-8B)  (↓) |
> > |-------------------------|--------|-------|-------|-------|
> > | NoWag-VQ col only       | 7.45   | 11.45 | 9.54  | 12.53 |
> > | NoWag-VQ row only       | 7.68   | 11.29 | 10.10 | 12.53 |
> > | NoWag-VQ row and column | **7.07**   | **10.68** | **9.12**  | **11.92** |
> >
> > **NoWag-P (50%) Ablation**
> > |                         |Wikit2 (2-7b) (↓) | Wiki2 (2-13b) (↓) | Wiki2 (3-8B)  (↓) | C4 (2-7b) (↓) | C4 (2-13b) (↓) | C4 (3-8B) (↓) |
> > |-------------------------|------|-------|-------|------|-------|-------|
> > | NoWag-P col only        | 7.01 | 5.53  | 9.26  | 9.22 | 7.44  | 10.60 |
> > | NoWag-P row only        | 6.42 | 5.82  | 8.49  | 8.37 | 7.88  | 9.78  |
> > | NoWag-VQ row and column | **6.37** | **5.49**  | **8.32**  | **9.12** | **7.35**  | **9.67**  |
> >
> > **Reason to Reject 4: Comparison with SparseGPT:**
> >
> > We compared the Wikitext2 Perplexity of NoWag against SparseGPT. SparseGPT The results are largely similar to the comparison against Wanda. Specifically, NoWag-P significantly outperforms SparseGPT at 50% pruning, and offers competitive performance at 4:8 and 2:4 semistructured pruning. In particular for the larger models (Llama 2 13B/70B and Llama-3 70B) NoWag outperforms SparseGPT for both 50%, 4:8 and 2:4 pruning.
> >
> > Wikitext2 PPL (↓)
> > |           | Sparsity |  2-7b | 2-13b | 2-70b |  3-8b | 3-70b |
> > |:---------:|:--------:|:-----:|:-----:|:-----:|:-----:|:-----:|
> > | SparseGPT |    50%   |  6.51 |  5.63 |  3.98 |  8.53 |  5.31 |
> > |  NoWag-P  |    50%   | **6.37**  |  **5.49** |  **3.89** |  **8.32** |  4.95** |
> > | SparseGPT |    4:8   |  **7.99** |  6.58 |  4.59 | **10.92** |  6.56 |
> > |  NoWag-P  |    4:8   |  8.04 |  **6.47** |  **4.45** | 12.66 |  **6.24** |
> > | SparseGPT |    2:4   | **10.23** |  8.29 |  5.38 | **14.31** |  8.62 |
> > |  NoWag-P  |    2:4   | 11.14 |  **8.28** |  **5.17** |  24.0 |  **7.52** |
> >
> > **Ethics Flag:** We noticed that you raised the Ethics flag in your review but did not notice anything in your review that would indicate why you raised the flag. We were hoping to ask what exactly about our paper prompted raising the Ethics flag.

---

> > > ### Comment · Reviewer_H7VY · 2025-06-02
> > >
> > > As the author has solved most of my concerns, I am willing to raise my score.
> > >
> > > PS: For the ethics flag, I think it is an accidental touch.

---

### Official Review · Reviewer_yN5X · 2025-05-13

**Rating:** 8
**Confidence:** 4
**Ethics Flag:** 1

**Summary:**

This paper studies the model compression. The paper clearly states the problem of model compression using calibrated datasets and the proposed approach. The main idea to to normalize the weight and then using the calibration dataset's activation as a weight to guide the VQ approach to compress the model weight more efficiently. The paper is well written and with a discussion section describing why the proposed approach works.

**Questions To Authors:**

This paper is so well and clearly written. It's a pleasure to read from the beginning to the end. Although I am not an expert in model compression, this paper is quite strong by providing both empirical results and discussions for describing why the proposed approach works. The figure 3 helps quite a lot for understanding the approach.

Just one quick question, how sensitive of the approach to the calibration dataset?

**Reasons To Accept:**

1. The paper is very well written and the problem statement is very clear.
2. The proposed approach makes a lot of sense and has illustration and explanation for why the proposed approach works.
3. The model achieves quite good performance for compressing the small size model.

**Reasons To Reject:**

None

---

> ### Author Response · Authors · 2025-06-02
>
> We sincerely thank Reviewer yN5X for the thoughtful and encouraging feedback. We're especially grateful for your positive remarks on the clarity of the paper and the usefulness of the illustrations, particularly Figure 3. We’re glad that our exposition made the proposed method accessible and enjoyable to read.
>
> **Regarding the question on sensitivity to the calibration dataset:**
>
> We ran an ablation on the NoWag-VQ dataset using 128 samples of the RedPajamas dataset and two other popular calibration datasets, the train split of the Wikitext2 and C4 datasets. Due to time constraints we only performed the ablation on the small models, Llama-2 7B and Llama-3 8B.  In essence the results regardless of calibration dataset is SOTA, however RedPajamas offers the best overall performance since it is the closest to the original pretraining dataset of the Llama models.
>
> **Wikitext 2 Validation Perplexity (↓)**
> | Calibration dataset | 2-7B | 3-8B  |
> |---------------------|------|-------|
> | Wikitext2           | **6.9**  | 10.9  |
> | C4                  | 7.19 | 11.22 |
> | RedPajamas          | 7.07 | **10.68** |
>
>
> **C4 Validation Perplexity (↓)**
> | Calibration dataset | 2-7B | 3-8B   |
> |---------------------|------|--------|
> | Wikitext2           | 9.16 | 12.55  |
> | C4                  | **9.12** | 12.055 |
> | RedPajamas          | **9.12** | **11.92**  |

---

> > ### Comment · Reviewer_yN5X · 2025-06-10
> >
> > Thanks for the responses. I maintain my ratings.

---

### Official Review · Reviewer_sxCa · 2025-05-25

**Rating:** 6
**Confidence:** 4
**Ethics Flag:** 1

**Summary:**

In this paper, the authors propose a model compression method, NoWag, that can be applie to both quantization and pruning. Unlike proor works that compress the original weight matrices, the authors propose to first normalize the weight matrices row-wise and column-wise, and then compress it. In this way, the elements in the weight matrix has similar scalar range that are friendly to compression. After the normalization, the weight matrix can be compressed by using the same methods proposed earlier, i.e. kmeans for VQ and Wanda for pruning.

After extensive experiments, the authos show:

1. Compared to VQ methods, NoWag outperforms QUIP#, and is comparable to AQLM.

2. Compared to Prunning method, NoWag outperforms Wanda for the 50% unstructured setting, comparable or mixed to Wanda for the 4:8 and 2:4 settings.

**Questions To Authors:**

See reasons to reject for clarification.

**Reasons To Accept:**

1. The proposed method is very easy to implement. Only one additional step is required prior to existing methods, i.e. normalization. And the overhead is neglegible.

2. The motivation for this normalization step is clear to me. It's similar to store the salient weight element in a two vector per-column and per-row.

3. The paper is clearly written, easy to follow.

**Reasons To Reject:**

1. **The results are a bit mixed**. For quantization, NoWag clearly outperforms QUIP#, but is comparable to AQLM. For pruning, NoWag clearly outperforms Wanda for the 50% setting, but is comparable for the 2:4 and 4:8 settings.

2. **Lack of baselines**. Inspired by the first step, I would suggest to include more VQ and Pruning baselines to show the benifit of NoWag. For example, VQ baselines could be GPTVQ [1] and ClusComp [2]. [2] seems a concurrent work, but a related discussion is also benificial. Pruning baselins could be DARE[3] and SLERP[4].

[1] GPTVQ: The Blessing of Dimensionality for LLM Quantization

[2] ClusComp: A Simple Paradigm for Model Compression and Efficient Finetuning

[3] Language Models are Super Mario: Absorbing Abilities from Homologous Models as a Free Lunch

[4] Spherical Linear Interpolation and Text-Anchoring for Zero-shot Composed Image Retrieval

---

> ### Author Response · Authors · 2025-06-02
>
> We appreciate Reviewer sxCa thorough review. Below we address the concerns raised in detail,
>
> **Reasons To Reject 1: The results are a bit mixed.**
>
> We respectfully disagree with the characterization that the results are too mixed to merit acceptance. While NoWag achieves performance comparable to AQLM and Wanda in some settings, it offers significant practical advantages that we believe warrant recognition.
>
> **Quantization:**
> Although NoWag matches AQLM in performance, it does so with dramatically reduced resource requirements. Specifically, NoWag uses 16× less calibration data and achieves quantization of a 7B model in 10 hours on a single A100 GPU—compared to AQLM, which requires over 1 day under the same conditions. These gains in efficiency are critical in real-world deployments, where compute and memory constraints are major bottlenecks.
>
> **Pruning:**
> It is important to note that structured pruning patterns like 2:4 and 4:8 inherently restrict the sparsity patterns we can induce, limiting the benefit of breaking up weight matrix structures that our normalizer provides. As expected, under more flexible patterns such as 8:16, 16:32, and 32:64, NoWag significantly outperforms Wanda (see Figure 4). Moreover, even at the constrained 2:4 setting, NoWag achieves competitive performance with Wanda while providing the additional benefit of using a metric that is compatible with quantization —something Wanda does not support. This unified applicability across compression paradigms is a key contribution of our method.
>
> **Reasons To Reject 2: Lack of baselines.**
>
> **Quantization**
> We conducted additional evaluations comparing the Wikitext2 (wiki2) and C4 perplexity between NoWag-VQ and ClusComp and GPTVQ. Perplexities were calculated with a context length of 2048, rather than the native context length of the models, since both ClusComp and GPTVQ reported perplexities calculated with a context length of 2048. GPTVQ did not report C4 perplexities.  For ClusComp we compared against ClusComp$^-$ the non fine tuned results, and ClusComp, the block wise fine tuned results.
>
> **One/Zero-shot**
> |              | bits  | Wiki2 (↓) 2-7B | Wiki2 (↓) 2-13B | Wiki2 (↓)  2-70B | C4 (↓)  2-7B | C4 (↓)  2-13B | C4 (↓)  2-70B |
> |--------------|-------|---------------------|----------------------|-----------------------|--------------|---------------|---------------|
> | GPTVQ        | 2.125 | 8.23                | 6.5                  | 4.64                  | -            | -             | -             |
> | ClusComp$^-$ | <2.01 | 52.38               | 22.9                 | 9.84                  | 50.08        | 24.47         | 13.96         |
> | NoWag-VQ     | <2.02 | **7.59**                | **6.37**                 | **4.41**                  |**9.28**         | **8.16**          | **6.34**          |
>
> **Blockwise Finetuned**
> |              | bits  | Wiki2 (↓) 2-7B | Wiki2 (↓) 2-13B | Wiki2 (↓)  2-70B | C4 (↓)  2-7B | C4 (↓)  2-13B | C4 (↓)  2-70B |
> |--------------|-------|----------------|-----------------|------------------|--------------|---------------|---------------|
> | ClusComp | <2.01 | 7.5            | 6.17            | 4.83             | 10.29        | 8.49          | 7.02          |
> | NoWag-VQ     | <2.02 |**7.01**           | **5.93**            | **4.25**             | **8.07**         | **7.64**          | **6.18**          |
>
> These evaluations show that **NoWag-VQ significantly outperforms both GPTVQ and ClusComp** in the respective non fine tuned and fine tuned settings. Furthermore, the non fine tuned NoWag-VQ performs competitively with the fine tuned ClusComp.
>
> **Differences with ClusComp:** There are two main differences between NoWag and ClusComp. First, ClusComp does not incorporate any normalization, unlike NoWag-VQ. Second, ClusComp performs vanilla vector K-means, rather than weighted K-means, meaning that ClusComp has no data awareness and treats all input channels with equal importance. This is why especially in the one-shot regime, ClusComp$^-$ performs so poorly compared to NoWag-VQ
>
> **Pruning**
>
> Respectfully, we don’t believe that the two baselines proposed are relevant baselines to compare against NoWag. DARE focuses on pruning to more effectively merge Supervised Finetuned LLMs on different tasks together. SLERP focuses on Composed Image Retrieval, we checked the main text of the paper and there is nothing related to pruning or quantization. Could you please clarify how SLERP and DARE are relevant papers to use as baselines to compare against NoWag-P?

---

> > ### Author Response · Authors · 2025-06-06
> >
> > Hello again and thank you for the time already invested in your review of our work!
> >
> > As there are only 4 days remaining in the discussion period, we wanted to follow up in case you had any questions or feedback we could address. Since some of the experiments may take a bit of time to run, an earlier response would be greatly appreciated so we can ensure we provide a meaningful reply in time. Thank you again for your review and engagement.

---

> ### Comment · Reviewer_sxCa · 2025-06-07
>
> Thank you for the detailed rebuttal. I'm convinced by the discussion, and willing to raise my score.
>
> About the pruning baselines, the authors are right. my referred baselines are more about merging baselines. You can ignore this part.

---

### Official Review · Reviewer_5R73 · 2025-05-27

**Rating:** 7
**Confidence:** 3
**Ethics Flag:** 1

**Summary:**

This paper introduces NoWag (Normalized Weight and Activation Guided Compression), a framework for post-training compression of Large Language Models (LLMs). The core idea is a two-step normalization process applied to weight matrices (first column-wise, then row-wise) followed by compression guided by the second moment of activations (specifically, diag(XX^T)). The authors propose two instantiations: NoWag-VQ for vector quantization using weighted K-means, and NoWag-P for un-/semi-structured pruning by removing elements with the smallest normalized weight magnitude multiplied by activation norms. Experiments on Llama-2 and Llama-3 models show that NoWag-VQ significantly outperforms SOTA zero-shot VQ methods like QuIP# (especially with much less calibration data), and NoWag-P is competitive with or slightly better than SOTA pruning methods like Wanda in terms of perplexity, while being comparable in zero-shot task accuracy. The paper argues that the proposed normalization makes the weight distribution more amenable to compression and reduces sensitivity to outliers.

**Questions To Authors:**

1. Could you please clarify the exact pruning score used for NoWag-P? Is it |W_bar_ij| ||X_j||_2 (as implied by line 134 and Fig 2 legend) or is it derived more directly from minimizing Eq. 3, e.g., by pruning based on (W_bar_ij)^2 ||X_j||_2^2? If it's the former, how does this align with the objective in Eq. 3 for pruning

2. (minor) The paper mentions (line 130) T << min(d_in, d_out) for the number of K-means iterations. For NoWag-VQ, 100 iterations were used (line 587). Could you comment on the sensitivity to T? Could fewer iterations still yield strong results, further speeding up the compression?

3. The computational complexity for NoWag-VQ is given as O(d_in d_out T 2^(n_bpv d)) (line 129). This is O(N_{total\_weights} T 2^{k}) where k is bits per subvector. Can you briefly compare this complexity to QuIP# or other VQ methods if readily available, beyond the Hessian-based methods which are O(d^3)? The main benefit seems to be fewer calibration samples rather than algorithmic complexity of the VQ step itself, correct?

**Reasons To Accept:**

1. **Clarity**: The paper is generally well-written and easy to follow. The core normalization idea is explained clearly, supported by algorithms and illustrative figures (Figs. 1, 2, 3) that effectively convey the intuition. The problem formulation and connection to existing work are well-established.

2. **Originality**: The main originality lies in the specific two-step normalization procedure (column-then-row normalization) and its application as a unified pre-processing step for both VQ and pruning, coupled with an activation-guided importance score. While components like weighted K-means or magnitude pruning with activation scaling are known, their combination with this particular normalization to improve compression is novel. The observation that this normalization creates a more "ball-shaped" distribution (Fig. 3) is an interesting insight.

3. **Significance**: The work is significant for several reasons: (1) It presents a simple yet effective technique that improves zero-shot VQ performance, notably outperforming SOTA (QuIP#) while using 48x less calibration data. This is a substantial practical contribution. (2) It offers a competitive pruning method (NoWag-P) that slightly improves upon Wanda in perplexity. (3) The analysis and visualizations provide valuable insights into how weight normalization can benefit LLM compression. It suggests a common underlying principle that can be beneficial for different compression paradigms, potentially inspiring future unified approaches.

**Reasons To Reject:**

1. "Unified Framework" Scope: The term "unified framework" is used, but the actual compression algorithms (K-means for VQ, greedy selection for pruning) remain distinct. The unification primarily comes from the shared normalization pre-processing step and the conceptual objective (Eq. 3). This is a minor wording concern rather than a fundamental flaw.

2. Performance on Llama-2 70B with Finetuning (Table 2): NoWag-VQ falls slightly short of AQLM on Llama-2 70B with finetuning. The authors attribute this to AQLM using d=8 vs. NoWag's d=7. While plausible, it's a point where NoWag-VQ isn't outperforming. This is a minor point as overall performance is strong.

3. Clarity on Pruning Score: There's a slight ambiguity in the pruning score formulation.
ore). This should be clarified: is the pruning score based on |W_bar_ij| ||X_j||_2 or (W_bar_ij)^2 ||X_j||_2^2 (or its square root)? The former is used in Wanda-like methods, the latter aligns more directly with minimizing the squared error in Eq. 3. The current text and Figure 2 legend imply the former (|W_bar_ij| ||X_j||_2). If so, the connection to Eq. 3 for pruning is less direct than for VQ.

---

> ### Author Response · Authors · 2025-06-02
>
> We thank the reviewers 5R73 for their thoughtful and constructive feedback. We are delighted that you found our paper clear and significant.
>
> **[Minor] Reason to Reject 1:  "Unified Framework" Scope:**
>
> Respectfully, we disagree with the assertion that the unified framework wording is not accurate. As you mentioned in your review, the “unified framework” refers to the normalization and conceptual objective. This allows for performance that is superior/comparable performance for pruning and compression using common, well explored, algorithms for each compression measure (K-means for VQ and greedy pruning for Pruning). However we remain open to suggestions for changes for more accurate wording.
>
> **[Minor] Reason to Reject 2: AQLM vs NoWag for Llama-2 70B.**
>
> We are planning on quantizing Llama-2 70B with a $d=8$ codebook for an apples to apples comparison with AQLM, however because the time taken to perform the quantization scales exponentially with $d$, with our computational resources it may not be finished before the end of the rebuttal period
>
>
> **Reason to Reject 3 and Question 1: Pruning Score**
>
> The pruning scores used by NoWag-P is $(\bar{W}_ij)^2 ||X_j||_2^2$. However we note that pruning based on $|\bar{W}_ij| ||X_j||_2$ is equivalent to pruning based on $(\bar{W}_ij)^2 ||X_j||_2^2$, as the latter is simply the square of the former, which is non-negative. As a result, the resulting pruning mask will be the same whether we prune against $|\bar{W}_ij| ||X_j||_2$ or against $(\bar{W}_ij)^2 ||X_j||_2^2$.
>
> **Question 2: Sensitivity to $T$ and less iterations**
>
> We conducted an ablation for $T=1$, $T=10$, $T=40$. Because of the time and computational constraints, we selected the smaller models, Llama-2-7b/13b and Llama-3 8B to conduct this ablation study. We listed the Wikitext2 and C4 perplexities below, **bolding** the smallest $T$ for which NoWag outperforms QuIP#. For Llama-2 7b and Llama-3 8B **NoWag with only one iteration of K-means was able to outperform QuIP#**, for Llama-2 13B, **after 10 iterations, NoWag-VQ was able to outperform QuIP#.** We also note that the performance of the model continues to increase as we increase the number of iterations.
> | Method      | Wikitext2 (2-7B) ↓ | Wikitext2 (2-13B) ↓ | Wikitext2 (3-8B) ↓ | C4 (2-7B) ↓ | C4 (2-13B) ↓ | C4 (3-8B) ↓ |
> |-------------|--------------------|---------------------|--------------------|-------------|--------------|-------------|
> | QuIP#       | 8.23               | 6.06                | 13.80              | 10.80       | 8.07         | 15.60       |
> | NoWag T=1   | **7.32**               | 6.20                | **12.44**              | **9.58**        | 8.36         | **13.96**       |
> | NoWag T=10  | 7.04               | **6.00**                | 11.08              | 9.19        | **8.03**         | 12.35       |
> | NoWag T=40  | 7.10               | 5.99                | 10.86              | 9.15        | 7.98         | 12.07       |
> | NoWag T=100 | 7.07               | 5.93                | 10.68              | 9.12        | 7.94         | 11.92       |
>
>
> **Question 3: Computational Complexity Compared with Other VQ algorithms**
>
> Yes you are correct that the overall computational complexity is $O(N_{\mathrm{TotalWeights}} T 2^{k})$. Because all VQ algorithms need to sweep over all the weights and all the vectors in the codebook, they all have a computational complexity of at least  $O(N_{\mathrm{TotalWeights}}  2^{k} C)$ where $C$ is the number of iterations of this sweep that is performed. QuIP#, VPTQ, GPTVQ etc have an additional overhead of inverting the hessian, which is $O(d_{in}^3)$ for each weight matrix. Assuming that on average $d_{in} \approx d_{out}$ for the whole model, this roughly translates to an additional overhead of $O(N_{\mathrm{TotalWeights}}^{3/2})$ overhead, In practice for $T=100$ we don’t see any significant speedup compared with QuIP#, we do see a roughly 2x speedup compared with AQLM. However compared to both QuIP# and AQLM we use over an order of magnitude less calibration data.

---

> > ### Comment · Reviewer_5R73 · 2025-06-07
> > **Thank you for your rebuttal**
> >
> > I have carefully reviewed all the authors' comments. Their responses are well-written and solve most of my concerns. I will keep my original ratings positive.

---

### Decision · Program_Chairs · 2025-07-08

**Decision:**

Accept

**Comment:**

This paper introduces NoWag (Normalized Weight and Activation Guided Compression), a framework for post-training compression of LLMs. The authors provided a strong rebuttal, which led to two reviewers increasing their scores. After the rebuttal, the paper received scores of 6678. All reviewers expressed positive opinions about the paper, highlighting several strengths: (1) The paper is very well written with a clear problem statement; (2) The proposed approach is well-motivated, with intuitive explanations and illustrations of why it works; (3) The method achieves strong performance in compressing smaller-scale models. Given these merits, the AC recommends accepting the paper.